

# Dynamic decoding and dual synthetic data for automatic correction of grammar in low-resource scenario

Ahmad Musyafa[1,2], Ying Gao[1], Aiman Solyman[3], Siraj Khan[4], Wentian Cai[1] and Muhammad Faizan Khan[1]

[1] School of Computer Science and Engineering, South China University of Technology, Guangzhou, China
[2] Department of Informatics Engineering, Pamulang University, South Tangerang, Indonesia
[3] Department of Computer Science, University of Milan, Milan, Italy
[4] School of Software Engineering, South China University of Technology, Guangzhou, China

## ABSTRACT

Grammar error correction systems are pivotal in the field of natural language processing (NLP), with a primary focus on identifying and correcting the grammatical integrity of written text. This is crucial for both language learning and formal communication. Recently, neural machine translation (NMT) has emerged as a promising approach in high demand. However, this approach faces significant challenges, particularly the scarcity of training data and the complexity of grammar error correction (GEC), especially for low-resource languages such as Indonesian. To address these challenges, we propose InSpelPoS, a confusion method that combines two synthetic data generation methods: the Inverted Spellchecker and Patterns+POS. Furthermore, we introduce an adapted seq2seq framework equipped with a dynamic decoding method and state-of-the-art Transformer-based neural language models to enhance the accuracy and efficiency of GEC. The dynamic decoding method is capable of navigating the complexities of GEC and correcting a wide range of errors, including contextual and grammatical errors. The proposed model leverages the contextual information of words and sentences to generate a corrected output. To assess the effectiveness of our proposed framework, we conducted experiments using synthetic data and compared its performance with existing GEC systems. The results demonstrate a significant improvement in the accuracy of Indonesian GEC compared to existing methods.

Corresponding author
Ying Gao, gaoying@scut.edu.cn

## INTRODUCTION

Grammatical error correction (GEC) is a natural language processing (NLP) task that aims to automatically correct spelling, grammar, and related errors in written text. GEC has been a very prominent NLP task in recent decades. Various approaches have been presented to improve the performance of the GEC system, ranging from rule-based (*Chauhan et al., 2023*), n-gram-based (*Li et al., 2022*), to state-of-the-art neural-based approaches such as statistical machine translation (SMT) (*Das et al., 2023*) and neural machine

translation (NMT) (*Solyman et al., 2022*; *Mahmoud et al., 2023*). The success of deep learning techniques such as NMT has helped advance GEC applications in both research and industry (*White & Rozovskaya, 2020*). The current deep neural network approach widely uses the encoder–decoder structure (*Schmaltz et al., 2016*). NMT has become a neural-based method that has been widely used by automated GEC systems. While the recent GEC system has been conceptualized as a machine translation task using the neural-based sequence-to-sequence (seq2seq) model over monolingual data (*Solyman et al., 2021*; *Stahlberg & Kumar, 2021*), the task is to correct or translate ungrammatical text into well-grammatical formatted text. In the context of neural-based GEC systems, the encoder encodes the incorrect input sequence (the source sentence), maps it into hidden vector representation, and the decoder decodes it into the correct output sequence (the target sentence).

This article introduces a GEC system for the low-resource and under-explored as well as underrepresented languages, with Indonesian as a case study. Indonesian is the Austronesian language branch from the Malay-Polynesian language family in the South East Asia Islands language system (*Ghosh, 2020*; *Lin et al., 2021*) and has the fourth largest speaking population in the world (*Koto, Lau & Baldwin, 2020*; *Aji et al., 2022*). However, one of the main factors contributing to the sluggish advancement of NLP applications for the Indonesian language is the lack of a substantial amount of accessible parallel training datasets and the complexity of Indonesian GEC, such as contextual errors, orthography errors, syntactic errors, grammatical errors, ambiguity, and various types of errors. To the best of our knowledge, there is no substantial parallel training dataset or tailor-made systems for the Indonesian GEC task, except for a small evaluation dataset prepared by *Lin et al. (2021)* and research conducted by *Musyafa et al. (2022)*. To overcome this obstacle, researchers have introduced techniques for data augmentation and synthetic data generation. *Sennrich, Haddow & Birch (2016a)* introduced the back-translation technique for NMT, which has been adapted for generating synthetic data in GEC. Since neural-GEC is a sequence-to-sequence task, similar approaches have been applied to generate synthetic data for GEC (*Stahlberg & Kumar, 2021*). The off-the-shelf attentive NMT technique was introduced by *Kasewa, Stenetorp & Riedel (2018)* to generate a synthetic error based on language-learners error distribution. *Takahashi, Katsumata & Komachi (2020)* investigated the effect of pseudo errors on learner sentences extracted from monolingual data using a data selection approach and realistic error injection. *Solyman et al. (2021)* developed the confusion function technique to generate synthetic data using a semi-supervised approach in Arabic GEC. On the other hand, there has been limited research on the Indonesian GEC system, with only a select few studies undertaking similar efforts. *Lin et al. (2021)* curated a small evaluation dataset using a classification-based method, while *Musyafa et al. (2022)* constructed a synthetic training dataset from a web-crawled monolingual corpus.

The above research on the generation of synthetic training data served as the basis for this effort, considering the great benefits contributed by synthetic data to improve the effectiveness of GEC performance. This work is inspired by *Stahlberg & Kumar (2021)*, which performed a comparison of the top two methods on the BEA19 shared task: the Inverted Spellchecker (*Grundkiewicz, Junczys-Dowmunt & Heafield, 2019*) and the

pattern+POS (part of speech) (*Choe et al., 2019*). We complement and combine modified versions of these two confusion methods to introduce a new data generation method. These two methods complement each other; the first method focuses more on spelling errors, while the second method focuses on context errors, grammar, *etc*. We hypothesize that these methods perform better at assembling confusion sets, a pivotal resource for evaluating advanced neural-based GEC models for low-resource languages. The utilization of both methods for generating confusion sets enhanced the ability of the GEC model during experiments. In addition, we propose an advanced neural GEC model based on Transformer-based (*Vaswani et al., 2017*) equipped with a dynamic decoding technique named IndoneSIAn GEC (SiaGEC). The dynamic decoding technique utilizes character and lexicon models during the decoding process so that the proposed model can produce grammatically correct text. Experimental results demonstrate that the SiaGEC model with the proposed dynamic decoding outperforms the baseline seq2seq model and previous models. In short, the article offers the following contributions:

- We introduce a SiaGEC model a Transformer-based system that employs a dynamic decoding approach to effectively address the complexity of the Indonesian GEC.
- We introduce InSpelPO a confusion method to generate synthetic training data to overcome the constraint of a lack of parallel training data in Indonesian GEC.
- We provide a synthetic parallel training dataset for Indonesian GEC that is available for open access. This resource has great potential for future Indonesian GEC research.
- We conducted experiments on the only available Indonesian benchmark and we reported precision, recall, F1 score, and BLEU-4 score. The results affirmed the efficacy of our proposed model, showcasing its superior performance compared to earlier models.

The article is structured as follows: We begin with a Background section, where we lay the foundation of our proposed approach, followed by a discussion on Related Works, highlighting the various GEC approaches that have been applied in the field. The Methodology section specifics our approach, detailing the methodologies and techniques employed. In the Experiments section, we outline the experimental setup, data, and procedures used to evaluate our method. The Result and Discussion section presents the outcomes of our experiments and provides an analysis of the findings. The article concludes with a Conclusion section, where we summarize our contributions and suggest directions for further research. For access to the code and supplementary files, please visit our GitHub repository (https://github.com/Almangiri/SiaGEC-framework).

# BACKGROUND

Transformer-based architecture has attracted much attention in NLP tasks such as language understanding, machine translation, and text summarization due to their ability to learn large amounts of textual data and also be able to capture contextual relationships between words and sentences. The architecture of this framework comprises an encoder and a decoder structure, both of which heavily rely on the mechanism of self-attention. In principle, the encoder–decoder structure of the seq2seq task in GEC aims to convert ungrammatical input text into grammatical output text (*Solyman et al., 2023*). The encoder

encodes the input text $X = (x_1, x_2, x_3, ..x_T)$, which has the potential for grammatical errors, into a fixed-length representation called a hidden vector $h_t = (h_1, h_2, h_3, ..h_T)$. Meanwhile, the decoder utilizes this representation $h_t$ to produce a target output sequence $Y = (y_1, y_2, y_3, ..y_{T'})$ (corrected sentence) one token at a time. The neural-based system calculates each token of encoder $x_T$ and decoder $y_{T'}$ using hidden vector representation $h_t$, like the following equation:

$$h_e^t = f_e(h_e^{t-1}, x_t), \tag{1}$$

where $f_e$ denotes the encoder representation function that can be applied by neural networks such as long-short-term memory (LSTM), recurrent neural network (RNN), convolutional neural network (CNN), and Transformer. $h_e^t$ denotes a hidden vector of the encoder and $h_e^{t-1}$ denotes a previous hidden vector, and it can be calculated as in the following equations:

$$h_d^{t'} = f_d(h_d^{t-1}, y_{t'-1}, c), \tag{2}$$

$$P(x|y) = g(h_d^{t'}, y_{t'}, c), \tag{3}$$

$f_d$ refers to the function of decoder representation, which is a neural network as in the encoder above, where $h_d^{t'}$ refers to the hidden vector of the decoder, $h_d^{t-1}$ refers to the previous hidden vector of the decoder. $y_{t'-1}$ denote to the previous generated token, and $C$ refers to the vector of context. While $p(x|y)$ is the probability of the seq2seq GEC model. In the GEC neural-based system, the training objective is to maximize the probability of training data, as in the following equation:

$$M = \sum_{(x,y)} log P(y|x). \tag{4}$$

This technique allows the GEC models to correct different types of errors including contextual, grammatical, and syntactical errors. However, Neural-based models have demonstrated superior performance to statistical and rule-based approaches, and have made significant strides in automatically correcting long pieces of text. In this context, we introduce GEC mode based on a Transformer-based model and dynamic decoding technique. This leverages contextual relationships between words and the surrounding sentences, employing an iterative process to generate contextually accurate corrections, which is inspired by NMT as proposed by *Meyer & Buys (2023)*.

## RELATED WORK

The enormous impact of successfully shared tasks (*Ng et al., 2014*; *Mohit et al., 2014*; *Lee et al., 2016*; *Bryant et al., 2019*) on the automatic GEC tasks has received great attention in industry, research, and real-world NLP technology development (*Solyman et al., 2021*; *Obied et al., 2021*). Various techniques and approaches have been offered and achieved promising results, ranging from classification techniques for specific types of error, rule-based systems, n-gram-based systems, and statistical machine translation (STM)-based

systems to advanced neural-based NMT techniques. The progress of the GEC task does not only occur in high-resource languages such as English and Chinese as universal languages (*Lin et al., 2021*), but also occurs in low-resource languages such as Malay, Arabic, Ukrainian, and Indonesian. However, the progression of Indonesian GEC as a low-resource language remains relatively sluggish due to the scarcity of accessible and organized parallel training data.

Regarding the problem of data scarcity in GEC tasks, many researchers have proposed various methods to generate synthetic parallel corpus in the last decade. *Felice et al. (2014)* developed a method that generates artificial sentence candidates that are applied to a hybrid approach, which is a fusion of a rule-based system and a system based on SMT. They obtained impressive results from the artificial data. *Sennrich, Haddow & Birch (2016a)* introduced a technique of back-translation, which translates target sentences into source sentences automatically, to generate synthetic source sentences from monolingual training data and they have reported substantial gains. *Xie et al. (2016)* introduced a recurrent neural network (RNN)-based approach to correct text, they trained the proposed model on additional synthetic error data including article or determiner errors and noun number errors using a data augmentation procedure. The same author presented a beam-search noising technique to generate a synthetic training set extracted from monolingual data and reported results similar to the original data (*Xie et al., 2018*).

In another study conducted by *Kasewa, Stenetorp & Riedel (2018)*, *Stahlberg & Byrne (2019)* and *Solyman, Wang & Tao (2019)*, they successfully applied the back-translation technique to the GEC task, which generates synthetic training data to learn the distribution of text errors and reported the result. Fluency-boost learning is an inference technique utilized to generate extra pairs of sentences during the training process proposed by *Ge, Wei & Zhou (2018)*. A spell checker technique, a method based on confusion sets that generate unsupervised synthetic errors (*Grundkiewicz, Junczys-Dowmunt & Heafield, 2019*). They used the synthetic data to train a GEC Transformer-based model, which was a functional and practical GEC system. *Lichtarge et al. (2019)* presented a round-trip translation, a technique that translates the source language to the target language and back to the source language. *Choe et al. (2019)* presented a realistic noising function, which is a morphological analysis method used to generate large erroneous data extracted from the training data and they reported the competitive result in the 2019-BEA shared task. Nonetheless, the above approaches are more frequently used in universal language machine translation tasks such as English and Chinese to increase the accuracy of practical systems such as GEC system (*Stahlberg & Kumar, 2021*).

In the past few years, several studies using synthetic data generation methods in low-resource languages have been conducted. *Náplava & Straka (2019)* provided synthetic GEC datasets in Czech, German, and Russian to train Transformer NMT models. They reported state-of-the-art results over the existing models. *Rothe et al. (2021)* proposed a language-agnostic technique to generate very large synthetic examples to train multilingual GEC models and achieve accuracy improvement across three low-resource languages. Two standard data generation techniques, spell-based and POS-based, were proposed by *Palma Gomez, Rozovskaya & Roth (2023)* for Ukrainian GEC. The synthetic data was used to train

the seq2seq Transformer model and scored as first on the GEC track and second on the GEC+Fluency track (where, which shared task, the name, and date are required). *Lin et al. (2023)* introduced a synthetic training data based on confusion sets for the Philippines Tagalog GEC system and gained competitive performance on the Tagalog corpus. In Arabic GEC, *Solyman et al. (2021)* introduces an unsupervised method to construct a large confusion sets-based synthetic data, which was used to train the SCUT-AGEC model. It was the first synthetic data used in the neural language model in Arabic GEC and reported encouraging results. *Solyman et al. (2022)* proposed a semi-supervised noising technique to produce synthetic training data and they achieved the best performance compared to the existing system.

To the best of our knowledge, there have been only a few studies that investigated Indonesian GEC to date. *Lin et al. (2021)* introduced LSTM based on an Indonesian GEC framework to correct ten types of Indonesian text-POS errors. They constructed a small synthetic data based on confusion sets to train the Transformer-based GEC model and achieved good performance. More recently, *Musyafa et al. (2022)* presented a semi-supervised confusion technique to generate artificial parallel data extracted from a monolingual dataset and they received significant results compared to the previous work. In short, the major distinction between the previous approaches and our proposed approach is that we combined two confusion sets-based synthetic generation methods, which were the two best methods on the shared task of BEA19 (*Grundkiewicz, Junczys-Dowmunt & Heafield, 2019*). Furthermore, we introduce a Transformer-based GEC system equipped with a dynamic decoding algorithm in the decoding process. The system is proven to be able to overcome the complexity of various types of grammatical errors in the Indonesian language.

## METHODOLOGY

In this section, we introduce our proposed framework in detail. This included an in-depth exploration of the proposed confusion method named InSpelPOS in two subsections (inverted spellchecker and patterns+POS). We also introduce a customized GEC model for low-resource languages based on a modified version of the Transformer-based equipped with a dynamic decoding technique. To tackle the bottleneck of training data, the constructed synthetic data plays a pivotal role in training and evaluating the prowess of our model. Furthermore, the proposed GEC model is tailored specifically for low-resource languages, for which we incorporate a dynamic decoding method that further refines our model's performance. The proposed SiaGEC framework in this article is illustrated in Fig. 1.

### Confusion method

The proposed confusion method is a combination of modified versions of two approaches. The first technique named Inverted-Spellchecker was introduced by *Grundkiewicz, Junczys-Dowmunt & Heafield (2019)* and the Patterns+POS method was proposed by *Choe et al. (2019)*, both approaches achieved significant improvements in GEC in English and other Latin languages. However, the constructed data was used to train the GEC model, which

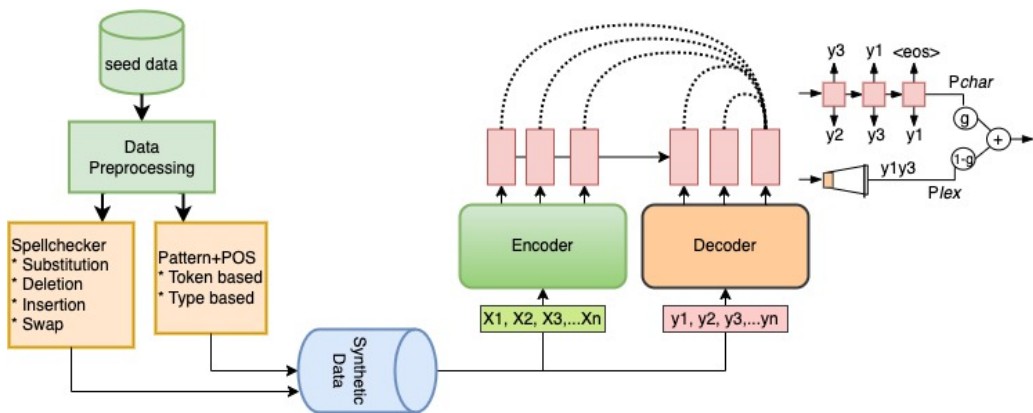

**Figure 1** Visual representation of the proposed SiaGEC framework.

left side of Fig. 1 provides a graphical representation of the process of generating synthetic data, and the following subsections provide detailed information.

### Inverted spellchecker

We implement a modified version of the Inverted-Spellchecker that was proposed by *Grundkiewicz, Junczys-Dowmunt & Heafield (2019)*. In order to generate a confusion set, we utilized the Aspell-spellchecker technique in the original article to generate a list of suggestions for given words. Subsequently, the list of suggestions is sorted by considering the weighted average of the proposed word edit distance and its phonetic equivalent distance. The system restricts the confusion set for the input word to the top 20 suggestions. The example of confusion sets generated by the spellchecker is shown in Table 1.

In contrast to the previous system (*Grundkiewicz, Junczys-Dowmunt & Heafield, 2019*), we made some modifications by changing the probability level of errors in system operation. To generate synthetic training examples, we execute the subsequent steps: Initially, the system infuses artificial errors into several words within the original error-free sentences sampled from a monolingual corpus of the web crawl data (*Wenzek et al., 2019*). Additionally, the system determines the number of words to modify in each sentence, drawing from the word error rate in the development dataset with a normal distribution average of 0.15 and a standard deviation of 0.2. Furthermore, each selected word is applied to one of the following operations: (1) substitution of an original word $w_i$ with a random word from the confusion set (with 0.2 probabilities), (2) deletion of an original word $w_i$ (with 0.2 probabilities), (3) insertion of $w_i$ with random words (with 0.2 probabilities), and (4) swapping $w_i$ with adjacent words (with 0.2 probabilities). In addition, to improve the model's ability to correct misspellings, we perform the above operation with the same probability at the character level at a rate of 0.1 in the source sentence to generate spelling errors. Examples of erroneous sentences, encompassing both synthetic and spelling errors, are displayed in Table 2.

**Table 1  Examples of spellchecker confusion outputs.**

| Source words | Confusion sets |
|---|---|
| short | sort, shut, source, shoot, sound, shirt, SO OUT, sold |
| paper | Papers, piper, pepper, peeper, pearl, peer, people |
| leave | Lift, LEave, left, LOVE, life, leaves, leaf, LIVE |
| and | end, AnDs, ON, ant, ands, ends, ads, in |
| talking | walking, tooling, whaling, taking, too king, looking |
| or | ORR, ore, orb, oral, oar, Ro, our |
| used | eased, issue, issued, sued, assumed, assured |

**Table 2  An artificial example of erroneous sentences, words in bold refer to incorrect words.**

| Input | Output |
|---|---|
| Original input | They have read many books and papers and have published many papers in international and highly reputable journals by collaborating with many researchers around the world. |
| Synthetic errors | They have **road** many books, **peeper end** have published **money** papers **on the** international and **height** reputable journals by a collaborating with **money** researchers around the **word**. |
| Spelling errors | They **hav** road many **boks**, peeper end published money papers on **teh** international and height **reputabele** journals by **collabroating** with money researchers around the **word**. |

### Pattens+POS

In the second part of InSpelPoS, we introduce the use of pattern and part-of-speech techniques. These techniques generate synthetic noisy data through two distinct strategies: token-based (pattern) and type-based (POS).

In the token-based strategy, we utilize a monolingual dataset derived from web crawl data to pinpoint error patterns. We perform text editing on the training data several times, typically not less than five. These edits simulate typical human errors, such as missing punctuation, verb tense inconsistencies, and prepositional errors. The output of these text edits is compiled into a dictionary that captures common edits made by human annotators on the in-domain training set. To generate realistic noise (synthetic errors), we apply this dictionary in reverse to grammatically correct sentences, forming a parallel corpus. In the type-based strategy, the system creates noise scenarios based on prior knowledge of the token type. We introduce noise based on POS, concentrating on three types of token errors: verbs, nouns, and prepositions, following the approach proposed by *Choe et al. (2019)*. Each token type is altered based on its common errors without changing its original type. For instance, prepositions are swapped with other prepositions, nouns switch between singular and plural forms, and verbs are replaced with their inflected forms.

In practice, we check if each token in the original training set is present in the edit pattern dictionary (first strategy). If it is not, we then resort to the type-based approach

(second strategy). This artificial data offers a diverse range of realistic noise scenarios, thereby boosting the effectiveness of the noising function.

## Dynamic decoding

The proposed SiaGEC framework uses dynamic decoding in the decoding process, which is an algorithm capable of simultaneously generating a token and computing the subsequent token probability by incorporating the lexicon and the character models. In this way, by incorporating both models, the algorithm dynamically adjusts subword segmentation during generation. The probability calculation for each sub-word is described in the following equation:

$$p(s_n|s_{<n},x) \approx p(s_n|\pi(s_{<n}),x) = p(s_n|y_{<m},x) \tag{5}$$

where $s_{<n}$ is a sequence of subwords that is converted by concatenation operator $\pi(s_{<n})$ into the raw character $y_{<m}$ of preceding subword $s_n$. The equation of $p(s_n|y_{<m},x)$ is based on incorporating the character and the lexicon as follows:

$$p(s_n|y_{<m},x) = g_m Pchar(s_n|y_{<m},x) + (1-g_m)Plex(s_n|y_{<m},x). \tag{6}$$

Equation (6) above states that the subword probability is obtained from the combination of character and lexicon probabilities.

In the dynamic decoding process, we model the subword boundary decision explicitly. For instance, when we create a character, we take into account the character potential on the left (previous) and the right (next) whether to continue or end the subword as done by a naive beam search (*Huang et al., 2019*). This aims to avoid bias towards short subword by not directly comparing complete and incomplete subwords. We have four considerations that can calculate the probability of the next character, as follows:

- **Co-End** (The left character continues a subword ended by the right character)
- **Co-Co** (Both left and right characters continue the same subword)
- **End-End** (Both left and right characters end the subword)
- **End-Co** (The character on the left concludes a subword while the one on the right continues it (starts a new subword))

We can perform the next-character probabilities and output one character at a time if we apply subword boundaries as follows $p_{co-end} > p_{co-co}$ or $p_{end-end} > p_{end-co}$. This is as described in Algorithm 1 below.

Essentially, the decoding algorithm will be more ideal if it makes final segmentation decisions based on the characters before and after the potential subword boundaries during the decoding process. Thus, all possible segmentation in each generation step can be considered properly. The four considerations that can compute the probability of the next character can be formulated as follows:

$$P_{end-end}(Y|y_{<m},x) = g_m Pchar(Y,<eos>|y_{<m},x) + (1-g_m)Plex(Y|y_{<m},x). \tag{7}$$

This is the first and simplest case, in which the generated previous character at position $m-1$ concludes a subword. Then, the single character $Y$ is the next subword probability.

---

**Algorithm 1** : The proposed dynamic decoding process

---

1: Input: $x \rightarrow$ *The source sentence*

2: Output: $y^* \rightarrow$ *The generated output a character sequence concluding with $< eos >$ (end of sentence)*

3: Notation: $C \rightarrow$ *A character vocabulary*

4: $y_{co}$ : *Partial output, last char ends subword*

5: $y_{end}$ : *Partial output, last char continues subword*

6: $Y_{co} = argmaxP_{end-co}(Y|X), Y_{co} = [Y_{co}]$

7: $\qquad Y \epsilon C$

8: $Y_{end} = argmaxP_{end-end}(Y|X), Y_{end} = [Y_{end}]$

9: $\qquad Y \epsilon C$

10: **while** $y_{end}[-1] \neq < eos >$ **do**

11: $\quad Y_{co-co} = argmaxP_{co-co}(Y|y_{co}, X)$

12: $\qquad Y \epsilon C$

13: $\quad Y_{end-co} = argmaxP_{end-co}(Y|y_{end}, X)$

14: $\qquad Y \epsilon C$

15: $\quad y_{co} = argmaxP(y)$

16: $\qquad y \epsilon [y_{co}, Y_{co-co}], [y_{end}, Y_{end-co}]$

17: $\quad Y_{co-end} = argmaxP_{co-end}(Y|y_{co}, X)$

18: $\qquad Y \epsilon C$

19: $\quad Y_{end-end} = argmaxP_{end-end}(Y|y_{end}, X)$

20: $\qquad Y \epsilon C$

21: $\quad y_{end} = argmaxP(y)$

22: $\qquad y \epsilon [y_{co}, Y_{co-end}], [y_{end}, Y_{end-end}]$

23: **end while**

24: $y^* = y_{end}$

25: **return** $y^*$

---

$< eos >$ is the special end of the subword token. In this context, Eq. (7) describes that all possible characters $Y$ in vocabulary can be computed and be candidate for the next character. The second case is the modification of the Eq. (7). The case is such that the character $m-1$ does not end a subword but represents the final character within the subword, as shown in the subsequent equation:

$$P_{co-end}(Y|y_{<m}, x) = g_m Pchar(Y, < eos > |y_{<l:m-1}, y_{<l}, x)$$
$$+ (1-g_m)Plex(Y|y_{<l:m-1}, y_{<l}, x) \qquad (8)$$

where $y_{<l:m-1}$ are the generated character in the current subword and $l$ is the current subword starting position that concluded at $m$. Equations (7) and (8) still provide candidates when the subsequent character ends a subword. To obtain the probability of the next character that starts and continues a subword, we modify Eq. (7), as the following:

$$P_{end-co}(Y|y_{<m}, x) = g_m Pchar(Y|y_{<m}, x) + (1-g_m) \sum_{s:s1=Y, s \neq Y} Plex(s|y_{<m}, x) \qquad (9)$$
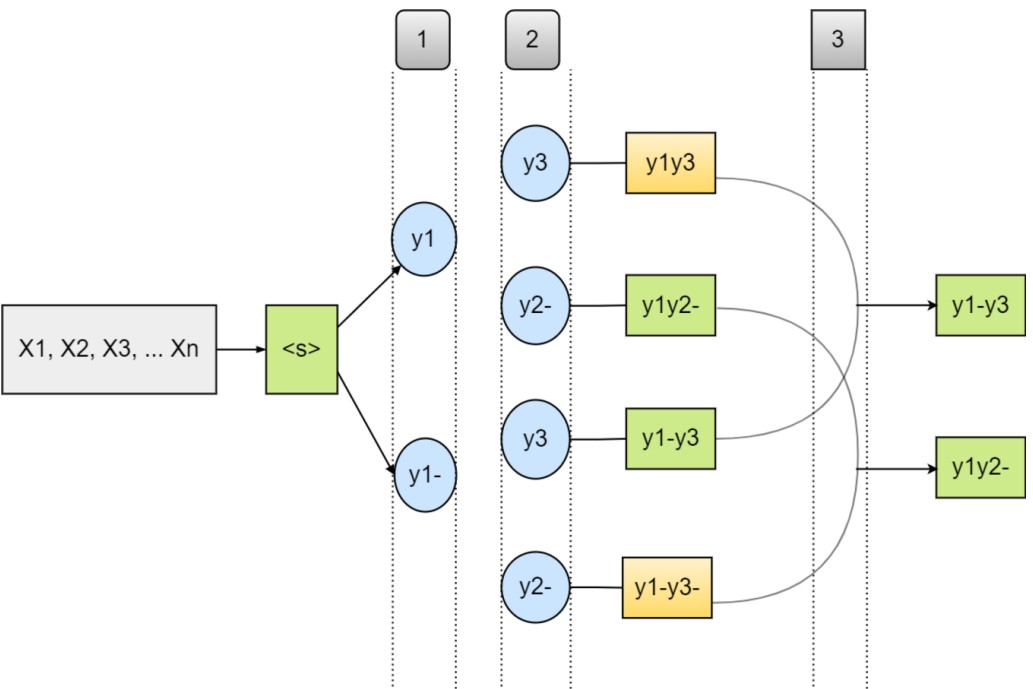

**Figure 2  An illustration of Dynamic decoding algorithm and the correction process on each character or token.** The '-' sign denotes sub-word boundaries: (1) illustrates the step of generating candidate characters that continue and end sub-words, (2) illustrates the steps that proceed one character forward, and (3) is a segmentation decision completion step. The sequences marked in green are selected before the ones in orange, owing to their higher sequence probabilities.

In Eq. (9) above, the probability of the succeeding character in the first component is exclusively dependent on the character-level model, excluding token ¡eos¿. Meanwhile, the other component omits all subwords that start with the specified character $y(s \neq Y)$, because it is a subword that ends with the character $m$. As a generalization of Eqs. (7) to (8), (9) can also generalize to the next equation, namely the case where the character $m$ continues the subword starting with the previous character. Thus, it produces the following equation; however, the dynamic decoding algorithm process is described in detail in Fig. 2.

$$P_{co-co}(Y|y_{<m},x) = g_m Pchar(Y|y_{<l:m-1},y_{<l},x)$$
$$+ (1-g_m) \sum_{s:s1=Y,s\neq Y} Plex(s|y_{<l:m-1},y_{<l},x). \tag{10}$$

Incorporating the Confusion Method, which combines the Inverted Spellchecker and Patterns+POS approaches, we generate synthetic data that enhances the training and evaluation of our GEC model. Dynamic Decoding, an integral part of our SiaGEC framework, further refines our model's performance by adjusting subword segmentation during generation. These methodological components collectively empower our framework to address the unique challenges posed by low-resource languages, showcasing its potential in the realm of Grammar Error Correction.

## EXPERIMENTS

In this section, we provide a comprehensive overview of the dataset employed to train our SiaGEC model, the data pre-processing workflow, model configuration, and model evaluation. To begin with, we construct a synthetic parallel training set using data from a monolingual web crawl (https://data.statmt.org/cc-100/). Subsequently, we conducted data pre-processing before model training. Our model's performance was assessed using the $F_1$ score and Bilingual Evaluation Understudy (BLEU) metrics. For more specific information, we refer to the following sub-sections.

### Dataset

In this work, we sourced the seed synthetic training data from CC100-Indonesian, a monolingual dataset that Facebook AI teams (_Conneau et al., 2020_) prepared. The data was collected from January–December 2018 using Commoncrawl snapshots in the CC-Net repository and organized into a single text document with a total data size of 36 GB. We selected the CC100 Indonesian corpus due to its open accessibility and its status as the most comprehensive monolingual Indonesian corpus available. This extensive dataset encompasses articles from a wide range of sources, covering diverse topics such as sports, health, economics, law, history, classic tales, and culinary recipes.

However, due to our hardware limitation, we utilized 711 MB from the entire CC100-Indonesian dataset. Following the data pre-processing phase, we employed the confusion method InSpelPoS as described in the Inverted Spellchecker subsection to generate a large synthetic training data totaling 3.9 GB (source and target). The comparison of data between original and synthetic data can be up to five times larger, meaning that the dataset used for training the proposed model (synthetic) is much larger than the original dataset. Finally, we divided the synthetic training data into two sets of categories consisting of a training set of 8,681,648 examples, and a valid set of 2,170,412 examples at 80% and 20%, respectively. The test set was a benchmark from the previous work consisting of 650,129 examples.

### Data preprocessing

In this context, the sustained progress of neural GEC is partly credited to the efficient handling of word representations and subword segmentation (_Meyer & Buys, 2023_). However, word-level techniques can give rise to numerous challenges beyond vocabulary limitations, whereas character-level methods tend to be computationally demanding and lack semantic richness (_Solyman et al., 2021_). To address the problem of open vocabulary, we utilized byte pairing coding (BPE) (_Sennrich, Haddow & Birch, 2016b_) algorithm to split and represent rare or unknown tokens into subwords (32K subwords) immediately after tokenizing the dataset with SpaCy (https://spacy.io/). We set the maximum input sequence length to 200 words for both the training and testing phases. This subword segmentation ensures consistency in tokens across training and development sets, ultimately enhancing model performance. In the inference phase, we manually select all the hyperparameters for our model based on our experience with similar models and considering the computational resources available.

## Model configuration

In the experiments, we trained the proposed model using the baseline sequence-to-sequence Transformer architecture (*Vaswani et al., 2017*). Besides adapting the decoder block with the dynamic decoding method as detailed in the methodology section, we also implemented several modifications to the baseline architecture, including adjustments to the model size, batch size, number of layers, and head attention. Initially, we adjusted the model size from 512 to 256 and modified the batch size from 2,048 tokens to 256. We then decreased the number of layers from six to four, while retaining the original number of head attentions to eight which plays a significant role in understanding the dependencies between different parts of the input sequence, thereby improving the model's ability to generate accurate and contextually relevant outputs. In order to boost performance, we adopted the learned positional coding technique rather than the static coding method from the original article (*Vaswani et al., 2017*). In line with the BERT configuration (*Devlin et al., 2019*), we refrained from using label smoothing. We utilized the static Adam optimizer (*Kingma & Ba, 2015*) with a steady learning rate of $1x10^3$, forgoing the warm-up and cool-down strategies. During training, we employed early stopping whenever there was no enhancement in validation data after 30 epochs. However, dropout was applied with probabilities of 0.1 and 0.15 to mitigate the risk of model overfitting. In our study, we intentionally set the hyperparameters manually. While automated methods for hyperparameter selection exist, we chose manual configuration during experiments for precise control over the models. All experiments were conducted using PyTorch in Python version 3.9 and executed on two NVIDIA Titan RTX GPUs with 25 GB of RAM each, making use of NVIDIA CUDA Toolkit 10.2.

## Evaluation

We applied MaxMatch scorer (*Dahlmeier & Ng, 2012*) as our evaluation metric to measure word-level edits within each output, subsequently providing precision, recall, and $F_1$ scores. $F_1$ scores aim to ascertain the precision, which attempts to calculate the percentage of accurate predictions to reveal the proportion of sentences corrected for grammatical errors. In this context, we calculate precision using the following equation, which indicates the percentage of errors that the system accurately corrected out of all the suggested corrections.

$$P = \frac{TP}{TP + FP} \tag{11}$$

where $TP$ represents true positives, indicating errors that the system has accurately corrected, whereas $FP$ denotes false positives, indicating errors that the system has not corrected. In this metric, a higher precision score signifies that the system corrects errors very well. On the other hand, we define recall as a good error detection technique, targeting the proportion of true positives to the sum of true positives and false negatives, as in the following equation:

$$R = \frac{TP}{TP + FN} \tag{12}$$

where *FN* represents false negatives, indicating the errors that the system did not detect. In this metric, a good recall score signifies that the system detects errors well. Finally, the $F_1$ score serves as a mean, striking a balance between precision and recall, ultimately indicating the system's proficiency in detecting and rectifying errors, as the following equation:

$$F_{1score} = \frac{2*P*R}{P+R}. \tag{13}$$

These metrics are able to assess various aspects of system performance, such as accurate identification and error correction of GEC systems, providing valuable perspectives on the system's strengths and areas for improvement.

Furthermore, we utilized the BLEU-4 score (*Papineni et al., 2002*) to measure the similarity between system prediction results and reference data. In the BLEU score, we used N-gram precision to compute the prediction result with reference data using a maximum size of four n-grams, as in the following equation:

$$BLEU - 4 = BP.exp\left(\sum_{n=1}^{4} w_n log p_n\right) \tag{14}$$

where $w_n$ denotes weight of n-gram (set to $\frac{1}{4}$) and $p_n$ represents precision. A higher BLEU score indicates greater similarity between the prediction results and reference data, with a score of 1 meaning the predicted and reference sentences are identical. Thus, BLEU scores tend to be high in GEC system evaluation. A brevity penalty (BP) yields a score of 1 when the prediction is overly short. Essentially, if the length of the predicted output exceeds the length of the reference data, the BP will produce a value of 1, as explained in Eq. (15).

$$BP = \begin{cases} 1 & if c > r \\ exp^{1-r/c} & otherwise \end{cases}. \tag{15}$$

The evaluation utilized a combination of precision, recall, and F1 score metrics to assess the system's ability to identify and correct grammatical errors. These metrics provided valuable insights into the system's strengths and areas for improvement. Additionally, we employed the BLEU-4 score to measure the similarity between SiaGEC predictions and reference data, offering a comprehensive evaluation of the system's overall performance. These evaluation methods together provide a well-rounded assessment of the GEC system's capabilities.

## RESULT AND DISCUSSION

This section presents the experimental findings from our SiaGEC model, which has been trained using synthetic datasets. Additionally, we have explored various seq2seq models, SiaGEC with dynamic decoding, multi-pass decoding, and a case study, which are elaborated upon in the subsequent subsections.

### Impact of the synthetic data
We investigated the effectiveness of InSpelPOS for generating dependable training data across various frameworks. Table 3 provides insights into the impact of synthetic data

**Table 3** Performance evaluation of GEC models for automated Indonesian text error correction including GEC-BRNN, GEC-CNN, and GEC-Trans (Vanilla) that were trained using our synthetic data (InSpelPOS), and two common State-of-the-Art confusion techniques: Inverse Spellchecker and Pattern POS.

| Model | Precision | Recall | F1 score | BLEU |
|---|---|---|---|---|
| GEC-BRNN + InSpelPOS | 56.20 | 43.70 | 49.17 | 62.07 |
| GEC-CNN + InSpelPOS | 66.23 | 51.18 | 57.74 | 72.00 |
| GEC-Trans + InSpelPOS | **72.09** | **64.89** | **68.30** | **74.33** |
| GEC-Trans + Inverse Spellchecker | 65.72 | 56.33 | 60.66 | 72.82 |
| GEC-Trans + Pattern-POS | 68.71 | 61.19 | 64.73 | 73.43 |

Notes.
Numbers in bold refer to the largest values.

over different architectures, including bidirectional recurrent neural network (BRNN), convolutional neural network (CNN), and self-attention network Transforemer-based (Vanilla) referred to as (GEC-Trans), as same as two common synthetic data generation techniques: Inverse Spellchecker *Grundkiewicz, Junczys-Dowmunt & Heafield (2019)*, and Pattern-POS *Choe et al. (2019)*. BRNN with an attention mechanism designed to focus on nearby words achieved an F1 score of 49.17 and a BLEU score of 62.07. Furthermore, GEC-CNN with an attention mechanism offers the advantage of parallel training without the need for sequential operations. This CNN architecture seamlessly merged feature extraction and classification into a single task, this approach led to an improvement in the F1 score (+8.57) and BLEU score (+9.93). We also investigated the baseline GEC-Trans, a model built upon a modified version of the Transformer architecture, allowing it to process the entire input sequence simultaneously, which achieved F1 score of 68.30 and BLEU score of 74.33. On the other hand, we investigate the reliability of InSpelPOS over two state-of-the-art confusion methods Inverted-Spellchecker and Pattern-POS. Inverted-Spellchecker aims to generate confusion sets considering only spelling errors which reported 60.66 and 72.82 for F1 score and BLUE scores, respectively. However, pattern and POS-based methods generate synthetic errors utilizing a dictionary of common edits from monolingual data, and introducing noise in verbs, nouns, and prepositions. This technique reported 64.73 F1 score and 73.43 BLUE score which is better as compared to Inverted-Spellchecker and behind our proposed InSpelPOS. These results underscore the effectiveness and reliability of the semi-supervised method for constructing training data in the context of Indonesian GEC and address the challenge of limited training data availability.

## Dynamic decoding

We evaluate the performance of SiaGEC using dynamic decoding and explore Word embedding, and various BPE settings, including 30k and 1,000 WordPiece units. Figure 3 shows that the incorporation of GEC-Trans with dynamic decoding and word embedding achieved F1 score of 70.93, marking an improvement of 2.63 points. Dynamic decoding enhances the model's capacity to comprehend and rectify errors within the broader sentence context, prioritizing critical errors while mitigating over-correction. We also investigated the impact of different BPE settings to address challenges related to unknown words. Initially, we adopted a 30k WordPiece unit configuration, consistent with previous work in

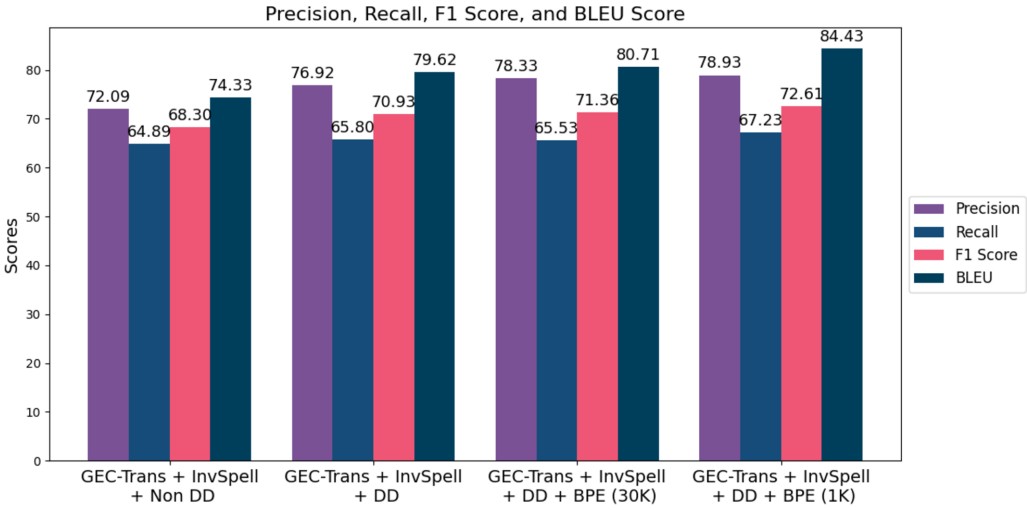

**Figure 3** The performance of SiaGEC for automatic correction of Indonesian errors is equipped with dynamic decoding (DD) and BPE variations (30k and 1k).

*Musyafa et al. (2022)*, resulting in performance gains of 71.36 for the F1 score and 80.71 for BLEU-4. Furthermore, we observed enhancements in the GEC-Trans + dynamic decoding model by employing a 1000 WordPiece unit BPE, raising the F1 score from 71.36 to 72.61 and the BLEU score from 80.71 to 84.43. Figure 3 presents the performance of SiaGEC with dynamic decoding under the 30k and 1k BPE settings.

## Multi-pass decoding

In the realm of correcting text fraught with multiple grammatical errors, using a single-pass inference approach proves to be a formidable challenge due to the intricate nature of human language. *Ge, Wei & Zhou (2018)* pioneered a multi-round correction method to enhance the fluency of GEC systems. This method entails iteratively refining a sentence, addressing its errors in multiple rounds, thereby progressively enhancing its fluency. In our SiaGEC model, we have integrated this approach. Initially, we trained a new version of SiaGEC from Right to Left (SiaGEC R2L) which receives an input sentence and produces an initial correction in a single round. Instead of treating this initial correction as the final output, we feed it into the (SiaGEC L2R) model to further refine the sentence in a multi-round fashion. This dynamic process results in our ultimate prediction, thereby capturing a more comprehensive correction path. In our experimental evaluation, we have established two baseline scenarios. In the first baseline, we follow the order of correction by first utilizing SiaGEC L2R and subsequently applying SiaGEC R2L. In the second baseline, we reverse the correction order, employing SiaGEC R2L before SiaGEC L2R. The results, as demonstrated in Table 4, indicate increases of 2.28 and 3.24 points in recall and 0.42 and 1.2 points in F-score for both baselines. However, it is worth noting that the precision exhibits an average decrease of 0.89 and 0.23 points in both scenarios. This decline in precision may be attributed to a potential imbalance issue between the two correction models (*Liu et al., 2016*).

**Table 4** SiaGEC model performance using multipass correction in two scenarios, numbers in bold refer to the largest values.

| Model | Precision | Recall | F1 score | BLEU-4 |
|---|---|---|---|---|
| SiaGEC L2R → SiaGEC R2L | 81.21 | 66.34 | 73.03 | 84.13 |
| SiaGEC R2L → SiaGEC L2R | **82.17** | **67.00** | **73.81** | **85.33** |

**Table 5** Performance of SiaGEC as compared to the existing Indonesian GEC systems, numbers in bold refer to the largest values.

| Model | Precision | Recall | F1 score | BLEU-4 |
|---|---|---|---|---|
| *Lin et al. (2021)* | 59.40 | 42.60 | 55.10 | N/A |
| *Musyafa et al. (2022)* | 71.14 | **72.76** | 71.94 | 78.13 |
| SiaGEC | **82.17** | 67.00 | **73.81** | **85.33** |

We evaluated the performance of the SiaGEC + multipass correction model by comparing it with recent Indonesian GEC systems. In this analysis, we considered several noteworthy systems. *Lin et al. (2021)* proposed an Indonesian GEC model that leveraged Part-of-Speech (POS) tags and was trained on a substantial dataset comprising 10,000 sentences, which successfully corrected 10 types of grammatical errors, achieving an F1 score of 55.10. Additionally, *Musyafa et al. (2022)* introduced a Transformer-based GEC system tailored for Indonesian, augmented with advanced copying mechanisms, reported 71.94 F1. Table 5, shows a comparison of precision, recall, F1, and BLEU scores, and clearly demonstrates that SiaGEC surpasses the performance of existing GEC systems.

## Case study

In this section, we evaluated how well the model performs using a case study with practical scenarios. The examples presented in this real-world scenario come from the Indonesian CC100 monolingual corpus. We showcased the output produced by various versions of SiaGEC, including the baseline Transformer-based model, the Transformer with dynamic decoding, and the Transformer with both dynamic and multi-pass decoding, as outlined in Table 6. Additionally, we provided the source, target, and English translation. The provided examples have several error categories as defined by *Musyafa et al. (2022)*, such as spelling errors in 1, 2, 5, 6, and 7. The syntactic or grammatical errors are in 8, 9, and 11. The word order errors are in numbers 3, 4, and 10. The punctuation error is in number 8. At the outset, the baseline model corrected all spelling errors that are in numbers 1, 2, 5, 6, and 7, but failed to correct word choice errors, punctuation errors, and grammatical errors that are in numbers 3, 4, 8, 9, 10, and 11. Then the second baseline (Transformer + dynamic decoding) corrected all spelling errors, grammatical errors, and several word orders, which are in numbers 1, 2, 4, 5, 6, 7, 9, and 11. However, the second model failed to correct errors in punctuation and word order, numbers 3, 8, and 10. Finally, the last model (Transformer + dynamic and multi-pass decoding) is able to correct all spelling, word order, and grammar errors but fails to correct one punctuation error, namely the comma in number 8. The good performance of our proposed model showed that it is able to identify and correct errors in Indonesian text. Meanwhile, the failure of our SiaGEC

**Table 6  Output of our SiaGEC model in several versions, words in bold refer to incorrect words.**

| Type | Example |
|---|---|
| Source | **Indonisya**[1(s)] , kepulauan terbesar **d**[2(s)] dunia, adalah rumah **populasi bagi**[3(o)] yang beragam dengan lebih dari 270 **penduduk juta**[4(o)] , memiliki warisan budaya yang **kya**[5(s)] dan mayoritas **muslin**[6(s)] , sambil juga memiliki pemandangan yang **menajubkan**[7(s)] termasuk,[8(p)] **pantai tropis-tropis**[9(g)] **hujan hutan**[10(o)] **lebat yang**[11(g)] . |
| Target | Indonesia, kepulauan terbesar di dunia, adalah rumah bagi populasi yang beragam dengan lebih dari 270 juta penduduk, memiliki warisan budaya yang kaya dan mayoritas Muslim, sambil juga memiliki pemandangan yang menakjubkan, termasuk pantai-pantai tropis dan hutan hujan yang lebat. |
| Translation | Indonesia, the world's largest archipelago, is home to a diverse population of over 270 million people, with a rich cultural heritage and a predominantly Muslim majority, while also boasting breathtaking landscapes, including tropical beaches and dense rainforests. |
| Trans. | Indonesia, kepulauan terbesar di dunia, adalah rumah **populasi bagi**[3(o)] yang beragam dengan lebih dari 270 **penduduk juta**[4(o)] , memiliki warisan budaya yang kaya dan mayoritas muslim, sambil juga memiliki pemandangan yang menakjubkan termasuk,[8(p)] **pantai tropis-tropis**[9(g)] dan **hujan hutan**[10(o)] **lebat yang**[11(g)] . |
| Trans. + Dynamic decoding | Indonesia, Kepulauan terbesar di dunia, adalah rumah **populasi bagi**[3(o)] yang beragam dengan lebih dari 270 juta penduduk, memiliki warisan budaya yang kaya dan mayoritas Muslim, sambil juga memiliki pemandangan yang menakjubkan termasuk,[8(p)] pantai tropis dan **hutan tropis**[10(o)] yang lebat. |
| Trans. + Dynamic decoding + Multi-pass decoding | Indonesia, kepulauan terbesar di dunia, adalah rumah bagi populasi yang beragam dengan lebih dari 270 juta penduduk, memiliki warisan budaya yang kaya dan mayoritas Muslim, sambil juga memiliki pemandangan yang menakjubkan termasuk,[8(p)] pantai tropis dan hutan hujan yang lebat. |

model to correct punctuation errors is not a big problem in Indonesian writing. SiaGEC is still acceptable and more effective than the first baseline model in the given example. Therefore, the proposed SiaGEC model has significant improvements over models from previous work.

## Discussion

The experiments have yielded promising results and insights, but it is important to acknowledge certain aspects where our findings may not fully capture the complexities of the low-resource GEC.

The experiments of this study have been focused on specific configurations, because the performance of machine learning models can be sensitive to hyperparameter choices, further exploration of hyperparameter tuning and sensitivity analysis is warranted to identify optimal settings for various scenarios and datasets. Furthermore, SiaGEC has demonstrated effectiveness in Indonesian GEC, but its generalizability to other languages may vary. Different languages exhibit unique grammatical structures, error patterns, and linguistic challenges. Extending SiaGEC to encompass a broader range of languages and language families would provide a more comprehensive understanding of the model's applicability. Moreover, the evaluations have primarily focused on the technical performance of the models. However, the long-term impact of language correction systems should also consider user experience, user feedback, and potential ethical implications.

On the other hand, future research should focus on incorporating real-world data to assess model performance in practical contexts. This can help validate the effectiveness of SiaGEC under more diverse and complex conditions. Furthermore, extending the model's capabilities to correct errors in multiple languages is an exciting avenue for future research. This involves accommodating variations in linguistic structures and error types across different languages. In addition, investigating the long-term impact, user experience, and ethical implications of language correction systems is crucial. This includes gathering user feedback and addressing any ethical concerns that may arise from automated correction. As well as a more in-depth analysis of specific error types and their correction can provide insights into areas where the model excels and where it requires improvement.

## CONCLUSIONS

This article introduced the GEC framework designed for low-resource languages applied to Indonesian as a case study. The framework incorporates a confusion method called InSpelPoS, which combines two techniques: Inverted spellchecker and Pattens+POS. This method has proven effective in generating a substantial amount of synthetic GEC data extracted from monolingual sources. The synthetic data generated by InSpelPoS addresses a significant challenge in GEC tasks for low-resource languages, such as Indonesian, where data scarcity is a common issue, and leveraging synthetic data aims to enhance the performance of GEC models. In addition, we introduce the concept of dynamic decoding within a Transformer-based model, specifically tailored for GEC tasks. Dynamic decoding methods have shown promising results in improving both precision and processing speed when identifying and rectifying errors in written language. This approach enables the utilization of advanced language models in GEC, facilitating a deeper understanding of context, meaning, and grammatical structure. As a result, it surpasses traditional rule-based techniques and those that operate within limited local contexts. Furthermore, exhibits superior competence in navigating the intricacies of natural language, encompassing aspects such as idioms, and cultural nuances, taking into account the broader context, as well as it can rectify a variety of error types, including grammar, sentence structure, and context. Additionally, the method's capacity to manage error chains, situations where a single correction necessitates multiple interconnected corrections within sentences, further sets it apart from traditional strategies.

In the future, the objectives include exercising control over synthetic error types through the utilization of error type tags for data augmentation. Additionally, we are keen on delving into neural-based methodologies that have the potential to expedite training while enhancing accuracy. Moreover, we plan to delve into the ramifications of SiaGEC on other GEC tasks, including text-to-speech and speech-to-speech applications.

### Funding

This work is supported by the Guangdong Provincial Key Laboratory of Artificial Intelligence in Medical Image Analysis and Application (No. 2022B1212010011). The funders had no role in study design, data collection and analysis, decision to publish, or preparation of the manuscript.

### Grant Disclosures

The following grant information was disclosed by the authors:
Guangdong Provincial Key Laboratory of Artificial Intelligence in Medical Image Analysis and Application: No. 2022B1212010011.

### Competing Interests

The authors declare there are no competing interests.

### Author Contributions

- Ahmad Musyafa conceived and designed the experiments, performed the experiments, analyzed the data, performed the computation work, prepared figures and/or tables, authored or reviewed drafts of the article, and approved the final draft.
- Ying Gao conceived and designed the experiments, authored or reviewed drafts of the article, and approved the final draft.
- Aiman Solyman conceived and designed the experiments, performed the experiments, analyzed the data, performed the computation work, authored or reviewed drafts of the article, and approved the final draft.
- Siraj Khan analyzed the data, prepared figures and/or tables, and approved the final draft.
- Wentian Cai analyzed the data, prepared figures and/or tables, and approved the final draft.
- Muhammad Faizan Khan analyzed the data, prepared figures and/or tables, and approved the final draft.

### Data Availability

The SiaGEC Framework is available at Zenodo: syafa_ahmad, & aimanmutasem. (2024). Almangiri/SiaGEC-framework: SiaGEC Release (v0.2.0). Zenodo. https://doi.org/10.5281/zenodo.10646584.

## Supplemental Information

Supplemental information for this article can be found online at http://dx.doi.org/10.7717/peerj-cs.2122#supplemental-information.

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
