# Peer review of "Dynamic decoding and dual synthetic data for automatic correction of grammar in low-resource scenario"

_PeerJ Computer Science, doi:10.7717/peerj-cs.2122_

## Round 0.1 · original submission · Major Revisions

The reviewer underline the relevance of this paper although it should be revised from different points of view.

In particular the authors should carefully take in account the suggestions provided by reviewer 1 and 3 in order to improve both methodology and evaluation section.

**Language Note:** PeerJ staff have identified that the English language needs to be improved. When you prepare your next revision, please either (i) have a colleague who is proficient in English and familiar with the subject matter review your manuscript, or (ii) contact a professional editing service to review your manuscript. PeerJ can provide language editing services - you can contact us at [email protected] for pricing (be sure to provide your manuscript number and title). – PeerJ Staff

Reviewer 1 ·

Basic reporting

1. The resolution of all the figures is low. Please provide clearer ones.

2. Redundant figures. Figure 2 is completely same as the left part of Figure 1, I think Figure 2 can be deleted. Figure 5 also has same contents as Table 3. One of them should be deleted.

3. Section of Pattens+POS is unclear. Both the two approaches are performed on verb and preposition, I can not understand the differences between them. More details need to be provided.

4. I think that the Section of Dynamic decoding follows the previous work Meyer and Buys (2023), please state it clearly.

5. The paper is about Indonesian but examples introduced in Tables 1 and 2 are English.

6.Missing section number in Line 260 and 281.

7. Typos: 'We' should be 'we' in line 13, redundant 'In' in line 122.

Experimental design

The experiments fail to support the main claim. In the Synthetic Data section, although the paper claims that Figure 4's results highlight the semi-supervised method's effectiveness in creating training data, the models differ solely in structure, indicating these results only demonstrate the structure's effectiveness. Similar issues arise in the Dynamic Decoding and Multi-Pass Decoding sections.

Validity of the findings

no comment

Reviewer 2 ·

Basic reporting

BASIC REPORTING
The manuscript present a confusion-based approach to generate synthetic data to be used in automatic correction of grammar structures, specifically tailored to low-resource and udner-represented languages. This domain is usually known as Grammar Error Correction (or GEC, for the sake of brevity). The case study revolves around the Indonesian language. The authors also provided a comparative analysis involving the proposed approach and other already existing GEC approaches.
The authors correctly started from the acknowledgement that whenever parallel corpora for NLP training are not available for a given language, the overall NLP advancement rate for that language is slower than for other languages.
The introductory section effectively supplies the reader with all needed elements to contextualize the research study. The research contributions are clearly stated at the end of the same section. I would encourage the authors to mention more explicitly in the fourth contribution, even if briefly, the different typologies of experiments performed during the validation stage on the available benchmark language corpora.
The proposed background section is rigorous and the proposed formulation of the problem helps to understand more clearly the research target. The provided core reference (i.e., Meyer and Buys) is up-to-date.
The related work section guarantees an adequate overview on the most recent studies on the addressed topic of GEC tasks and systems.
The section dedicated to the adopted methods is well-written and the proposed mathematical formulation is advisable and clear. Tables 1 and 2 are especially useful to the reader in order to understand the meaning of spellchecker confusion sets and the different typologies of erroneous sentences considered.
The experiments section is adequate. I suggest the authors to provide more details about the testbed they used (in terms of hardware and software specifications) so to motivate the use of a subset of the original CC100-Indonesian dataset. The model configuration should be motivated more explicitly. In other words, I would have expected the authors to explain why they opted for the selected values of a) model size, b) batch size, c) number of layers, and why they kept the original number of head attentions. I also encourage them to explain why the head attentions is relevant parameter to be considered when using the seq2seq transformer architecture (this could be placed, for instance, in the background section, where the transformer-based architecture is introduced and explained).
The section dedicated to the achieved results is well-described and plenty of details for the reader. Similarly, the discussion of the achieved results in the case study is well-supported by the example in Table 6. However, the identification of the different error types (highlighted in red in Table 6) and described in the text of the “Case Study” section, is less readable than the rest of the paper. I suggest the authors to consider whether adding another table where the different error categories are listed. The same table could also report, in additional columns, the behaviour of the various evaluated versions of SiaGEC (i.e., baseline Transformer-based model, transformer with dynamic decoding, transformer with dynamic decoding and multi-pass decoding) against the various error categories, in order to show also schematically how those error categories are tackled by SiaGEC versions.
The overall language used in the manuscript is adequate and no significant errors have been identified.

Graphical layout and figures: some figures have a low in-text resolution (i.e., Fig. 1, Fig. 3). On page 13, L265, there is numerical quantity written in red that should be changed into black font colour.

Experimental design

I have nothing to add since I noticed that:
1) the primary research falls within the journal's aims and scope
2) the research contributions are well-defined, relevant, and meaningful. Similarly, the research gaps to be filled are clearly stated
3) a rigorous investigation was performed
4) methods adopted are described adequately

Validity of the findings

I have nothing to add since I noticed that:
1) the work impact is correctly assessed, the replication of the study is encouraged (especially for other low-resource and under-represented languages)
2) data have been provided with an adequate level of details and correctness
3) conclusions are well stated

Reviewer 3 ·

Basic reporting

Writing can be significantly improved, particularly in more precisely specifying the original contributions of the paper and more clearly substantiating those contributions (see detailed comments on Experimental design below).

Minor comments:
- The authors should double-check what the appropriate citation style for this journal is. From what I can tell based on browsing other papers (e.g. https://peerj.com/articles/cs-1224/#intro), it is author-year. The manuscript uses only “author (year)” citations throughout, and no parenthetical citations “(author, year)”. The former should only be used when the cited work forms part of the text, e.g. “author (year) proposes method X” and the latter otherwise, e.g. “we method differs from method X (author, year)”. Using only the former style throughout makes it harder to read sentences because one expects a complete sentence to be formed in such cases.

- From what I can tell this journal does not use numbered sections. I suggest finding a more appropriate way to provide the overview (Lines 84-89). Additionally, there are several missing references to section numbers in the text, e.g. Line 281 “… as detailed in Section, we also implemented…”. If the journal provides no way to use numerical section headings, then I suggest using the title of the relevant section because it is otherwise hard to quickly flip back and forth to the referenced sections.

- It is noted (Lines 79-80) that the synthetic parallel dataset is available for open access. Is this the “Train set” proved as a Google Drive link in the GitHub repository? As with all low-resource work, the dataset is a vital contribution and I suggest finding a more citable, stable storage option (e.g. Zenodo) if PeerJ does not already provide this option.

- The word “Illustrative” should be removed from Figures 4 and 5. Using “illustrative” in this context suggests the numbers were made up for illustrative purposes, which I assume is not the case!

Experimental design

Based on the title and abstract of the paper, my expectations as a reader are that the two original contributions of this paper are 1) a synthetic data generation method that combines two previous approaches that offer complementary benefits, and 2) a dynamic decoding method that addresses shortcomings of existing decoding methods. If these expectations are correct, then the way the experiments and results are presented could be revised to better meet those expectations that have been set up (or different, more appropriate expectations should be set up for the reader early in the paper). Here are main ways in which the current presentation led to some confusion:

1. The authors appropriately cite Musyafa et al. (2022), one of the authors’ previous papers, which also proposes a Transformer-based model for Indonesian GEC. Is the main contribution of this paper the addition of the dynamic decoding? If other improvements to the architecture are proposed (i.e. not related to the decoding process), they should be made clearer. In this way, Figure 4 is confusing because the (repeated?) comparisons between GEC-BRNN vs. GEC-CNN vs. GEC-Trans. If a Transformer-based approach has been previously established as being superior (e.g. in Musyafa et al., 2022), then what motivates these current comparisons? If the current GEC-Trans. improves upon Musyafa et al. (2022), then what changes led to improvements, and where are the ablations showing which changes yielded the best improvements? If such architectural comparisons are not the focus of this paper (which only relate to synthetic data and dynamic decoding), then simply state the changes in the “Model configuration” section, and the results should focus only relevant comparisons.

2. The "Synthetic data" subsection (Lines 313-330) does not appear to present improvements resulting from removing/changing synthetic data generation methods but rather the architectural changes (e.g. BRNN vs. CNN vs. Trans.). If the content of the paragraph is as intended, then the title should be changed to reflect this. However, since the title and abstract of the paper suggest that InSpelPoS is a novel approach (which combines two existing approaches, Inverted Spellchecker and Patterns+POS), I would expect there to be a comparison of baseline approaches with only Inverted Spellchecker or Patterns+POS, and then a demonstration that the combination yields better results than either one alone (while keeping the Transformer architecture constant). If it is already well established that Inverted Spellchecker and Patterns+POS are complementary, there should be appropriate citations and, more importantly, perhaps adjust the emphasis of the title/abstract to indicate that the dynamic decoding is the primary, novel contribution.

3. Similarly, in the “Dynamic decoding” subsection, Figure 5 and Table 3 do not appear to present improvements from the addition of dynamic decoding, since each result in Figure 5/Table 3 includes dynamic decoding. Further, in line 334 there is a reference to Table 3 and F1 scores of 68.30 vs. 70.93, but the 68.30 does not appear anywhere in Table 3. Presumably this 68.30 is the GEC-Trans. without dynamic decoding. If such non-dynamic vs. dynamic decoding comparisons have already been completed, then they should be included in Figure 5/Table 3 to clearly demonstrate the benefits of dynamic decoding before discussing how various hyper-parameters related to dynamic decoding affect task performance.

Validity of the findings

No comments

Additional comments

I think the paper is interesting: it releases a dataset in a low-resource setting and provides a sound, novel decoding method for GEC. However, I think the paper could be made stronger by revising the presentation of the results to better fit the earlier sections of the paper (or revise those earlier sections to better introduce what is contained in the results). Good luck!

---

## Round 0.2 · Minor Revisions

The authors have submitted a revised version of their paper. The authors should revise the paper according the comment of reviewer 1.

Reviewer 1 ·

Basic reporting

This version is better compared with previous one.

Experimental design

In Table 3, could you please provide the result from vanilla GEC-Trans? This can explain the effectivenss of Inverse Spellchecker and Pattern POS.

Validity of the findings

none

Additional comments

none

Reviewer 2 ·

Basic reporting

The authors addressed all the comments reported in my previous round of review and no other interventions are required in my opinion.

Experimental design

The authors addressed all the comments reported in my previous round of review and no other interventions are required in my opinion.

Validity of the findings

The authors addressed all the comments reported in my previous round of review and no other interventions are required in my opinion.

Additional comments

The authors addressed all the comments reported in my previous round of review and no other interventions are required in my opinion.

Reviewer 3 ·

Basic reporting

I thank the authors for their efforts in addressing my comments raised in the first round of review. I think this version of the manuscript reads substantially clearer and I have no further comments.

Experimental design

No comment

Validity of the findings

No comment

---

## Round 0.3 · accepted · Accept

The authors have revised the manuscript according to the reviewers suggestion. I'm pleased to inform you that I have accepted your paper for publication.